# Evaluation of Pre-Therapeutic Assessment in Endometrial Cancer Staging

**DOI:** 10.3390/diagnostics10121045

**Published:** 2020-12-04

**Authors:** Caroline Bouche, Manuel Gomes David, Julia Salleron, Philippe Rauch, Léa Leufflen, Julie Buhler, Frédéric Marchal

**Affiliations:** 1Surgery Department, Institut de Cancérologie de Lorraine, 6, Avenue de Bourgogne, 54519 Vandoeuvre-les-Nancy, France; bouche.caroline31@gmail.com (C.B.); m.gomesdavid@nancy.unicancer.fr (M.G.D.); p.rauch@nancy.unicancer.fr (P.R.); l.leufflen@nancy.unicancer.fr (L.L.); f.marchal@nancy.unicancer.fr (F.M.); 2Biostatistics Department, Institut de Cancérologie de Lorraine, CEDEX 54519 Vandoeuvre-les-Nancy, France; j.salleron@nancy.unicancer.fr; 3CRAN, UMR 7039, CNRS, Université de Lorraine, Boulevard des Aiguillettes, 54506 Vandoeuvre-les-Nancy, France

**Keywords:** endometrial cancer, MRI, pre-therapeutic staging

## Abstract

Objective: The aim of this retrospective cohort study is to evaluate the concordance between the preoperative MRI and histology data with the final histopathological examination. Method: This is a retrospective observational study of 183 patients operated for endometrioid cancer between January 2009 and December 2019 in the surgical oncology department of the Lorraine Cancer Institute (ICL) in Vandœuvre-lès-Nancy. The patients included are all women operated on for endometrioid-type endometrial cancer over this period. The exclusion criteria are patients for whom the pre-therapy check-up does not include pelvic MRI and those who have not had first-line surgery. The final anatomopathological results were compared with preoperative imaging data and with endometrial biopsy data. Results: For the myometrial infiltration, the sensitivity of MRI was of 37% and the specificity of 54%. To detect nodal metastases, the sensitivity of MRI was of 21% and the specificity of 93%. We observed an under estimation of the FIGO classification (*p* = 0.001) with the MRI in 42.7% of cases (n = 76) and an overestimation in 24.2% of cases (n = 43). There was a concordance in 33.1% of cases (n = 59). We had a poor agreement between the MRI and final histopathological examination with an adjusted kappa (κ) of 0.12 [95% IC (0.02; 0.24)]. There was a moderate concordance on the grade between the pretherapeutic biopsy and the final histopathological examination on excised tissue with an adjusted kappa of 0.52 [95% IC 0.42–0.62)]. Endometrial biopsy underestimated the tumor grade in 28.9% of cases (n = 50) (*p* < 0.001), overestimated the tumor grade in 6.9% of cases (n = 12) and we observed a concordance in 64.2% of cases (n = 111). Conclusion: The pre-operative assessment of endometrial cancer is inconsistent with the results obtained on final histopathological examination. A study with a systematic review should be done to assess the performance of MRI, only in expert centers, in order to consider a a specific care management for endometrial cancer patients: patients who have had an MRI in an outpatient center should have their imaging systematically reviewed, with the possibility of a new examination in case of incomplete sequences, by expert radiologists, and discussed in multidisciplinary concertation meeting in expert centers, before any therapeutic decision. The sentinel node biopsy must be used for low and intermediate risk endometrial cancer.

## 1. Introduction

Endometrial cancer (EC) is the most common gynecologic cancer in France with 8220 new cases in 2018, ranking fourth among cancers in women. The overall survival is 75% at 5 years and 68% at 10 years owing to an early diagnosis permitted by vaginal bleedings. This prognosis and the risk of recurrence is linked to lymph node infiltration with an overall survival at 5 years of 89.6% at stage I compared with 49.4% at stage IIIC [1,2].

The lymph node evaluation depends on magnetic resonance imaging (MRI) which determines the International Federation of Gyneacology and Obstetrics (FIGO) classification. The stage depends on the myometrial invasion, cervix invasion, serous invasion, annexial and parametrial infiltration, lymphatic node infiltration, and bladder or intestinal or metastasis infiltration (stage I to IV) [3]. Metastatic lymph node involvement depends on myometrial infiltration and the degree of histological differentiation of the tumor, ranging from 1% for well-differentiated tumors limited to the endometrium, to 36% for undifferentiated tumors with more than 50% myometrial infiltration [4,5,6]).

Management of women with EC is hysterectomy with annexectomy for every stage. Indication of pelvic and para-aortic lymphadenectomy depends on recurrence risk defined by histological grade and FIGO stage. There are two types of EC: type 1 EC corresponding to adenocarcinomas and type 2 corresponding to clear cell carcinomas, papillary serous carcinomas, and carcinosarcomas. For type 1 EC, lymphadenectomy is recommended since stage II on FIGO’s classification. For early stages (IA and IB), a risk estimation based on histological grade and lympho-vascular space invasion (LVSI) on endometrial biopsy determines the indication of lymphadenectomy. Lymphadenectomy is not recommended for low risk EC, optional for intermediate risk EC and recommended for high risk EC according to European Society for Medical Oncology (ESMO) in 2016 [7] and the National Institute of Cancer (InCa) in 2010 [8]. Lymphadenectomy is always recommended for type 2 EC.

Lymphadenectomy allows the diagnosis of lymph node metastases, indicating the need for adjuvant treatment. This adjuvant treatment leads to benefits for patient’s survival [9,10], except for early stages with a low risk of recurrence [11,12]. However, pelvic and para-aortic lymphadenectomy are associated with high morbidity with hemorrhagic complications, lymphocele, lymphoedema, and nerve damage. Para-aortic lymphadenectomy is the main source of complications. The complication rate is 8% in case of initial lymph node dissection and 22% in the case of revision surgery [13].

Several studies showed the poor performance on MRI for the identification of lymph node disease [14,15,16]. Other studies demonstrated that the combined evaluation with MRI and endometrial biopsy also have a moderate sensitivity and specificity [17].

Preoperative assessment of the risk of lymph node metastasis is critical in the management of Type I endometrial cancer. In fact, it is a prerequisite for pelvic and para-aortic lymphadenectomy. The aim of this retrospective cohort study is to evaluate the concordance between the preoperative MRI and histology data with the final histopathological examination.

## 2. Material and Method

### 2.1. Population

We conducted a retrospective observational study of 183 patients retrieved from the systematic prospective clinical database registry of the Lorraine Cancer Institute (ICL) in Vandœuvre-lès-Nancy.

The study included women operated on for endometrioid cancer between January 2009 and December 2019 in the surgical oncology department.

The patients included were all women operated on for endometrioid-type endometrial cancer without neoadjuvant therapy over this period. Inclusion of patients was based on the histological diagnosis of endometrioid in pre- or post-operative histopathological examination. The exclusion criteria were: patients for whom the pre-therapy check-up did not include pelvic MRI and those who have not had first-line surgery. The pathologic diagnosis of endometrial cancer was made preoperatively by the Pipelle de Cornier^®^ biopsy, curetage hysteroscopy, or other sampling such as a polyp delivered through the cervix. All records were analyzed using PICSEL software to study the stage of the pre-operative MRI staging, the presence of LVSI, and grade on histological analysis. The data collected were: clinical patient’s characteristics (age, body mass index, menopausal status, hormone replacement therapy, tamoxifen use, parity, American Society of Anesthesiologists score and co-morbidities), postmenopausal metrorrhagia, pre-operative MRI analysis, pre- and post-operative histological data of the tumor, and surgical treatment modalities were noted. Staging was performed according to the FIGO 2009 classification. The ASA (American Society of Anesthesiologists) score qualifies a patient’s preoperative health status. It evaluates the anesthetic risk, morbidity (postoperative infection, infarction, respiratory or renal failure …), and mortality. There are six classes. ASA 1 and 2 group patients who are generally in good health. ASA 3 and 4 include patients with serious pathologies. Class 5 includes moribund patients and class 6 includes brain-dead patients. Staging was performed according to the FIGO 2009 classification.

For the imaging data, we focused on myometrial infiltration less than or greater than 50%, cervico-isthmic extension, ectopic infiltration, and lymph node infiltration. We identified the presence of fibroma, adenomyosis, endometriosis, myometrial atrophy, extension of the lesion into the uterine horn, the presence of myomas or adenomyosis. MRI were more often performed in external centers than in the expert center. The MRI protocol includes a tracking sequence followed by an axial and sagittal sequence with T2 fast spin echo (FSE) weighting. In case of uncertainty on T2-weighted images, MRI protocol was extended to dynamic sagittal T1-weighted sequences. The orientation of the dynamic images was parallel to the major uterine axis which corresponds to the maximum length of the endometrial cavity. T1-weighted axial images were obtained from the pubic symphysis to the renal hilum to identify the pelvic or para-aortic lymph nodes. MRI images were reviewed by only one radiologist.

For pathology data, we were interested in the type of preoperative specimen, histological type, tumor grade, and the presence of LVSI in pre- and post-operative patients. The histopathological examination was carried out in external centers or in the expert center and all the final histopathological examinations were carried out in the expert center.

The final anatomopathological results were compared with preoperative imaging data and with endometrial biopsy data.

This study was reported to the CNIL (French data protection agency) and approved by the institutional ethics committee of Lorraine Cancer Institute.

### 2.2. Statistical Method

Quantitative parameters were described as median, interquartile range (IQR), and qualitative parameters as frequency and percentage. All data were used for statistical analysis when available. The normality of the distribution was assessed with the Kolmogorov test.

Agreement between pre- and post-operative data was investigated with Kappa coefficient with 95% confidence interval. A value less than 0.4 was considered as poor agreement and a value from 0.4 to 0.6 as moderate. Over or under estimation between the two results was assessed with Wilcoxon test or Mac Nemar test. Sensitivity and specificity were computed considering histopathological examination as the gold standard.

Patients with and without an agreement between MRI and histopathological examination were compared with Chi-square test or Fisher exact for qualitative parameters and with the Mann Whitney U test for quantitative ones.

Statistical significance was set 0.05. Statistical analyses were performed with SAS software, version 9.2 (SAS Institute Inc., Cary, NC, USA).

## 3. Results

### 3.1. Population 

One hundred and eighty-three women underwent surgery without neoadjuvant therapy for an endometrioid carcinoma between 1 January 2009 and 31 December 2019.

One hundred and seventy-eight women had an estimation of FIGO stage on pre- and post-operative data and 173 had an estimation of tumor grade on pre- and post-operative data.

Median age of diagnosis was 66 years old and median BMI was 39 kg/m^2^. One hundred and seventy-six women (96.7%) were postmenopausal and 28 (18.1%) used a hormonal replacement therapy. Twelve women (6.6%) had used tamoxifen. For diagnosis, 166 (90.7%) women had abnormal bleedings at diagnosis (Table 1).

### 3.2. Concordance between MRI and Final Histopathological Examination 

For the myometrial infiltration, the sensitivity of MRI was of 37% and the specificity of 54%.

To detect nodal metastases, the sensitivity of MRI was of 21% and the specificity of 93%.

The MRI underestimated the FIGO stage in 42.7 % of cases (n = 76) (*p* = 0.001) and overestimated the FIGO stage in 24.2% of cases (n = 43). There was a concordance in 33.1% of cases (n = 59). We had a poor agreement between MRI and final histopathological examination with an adjusted kappa (κ) of 0.12 [95% IC (0.02; 0.24)] (Table 2).

Women with a concordance between MRI and final histopathological examination on FIGO stage had a median BMI of 35.3 kg/m^2^ compared to 39.6 kg/m^2^ for women with a discordance (*p* = 0.05). Focusing on myometrial invasion, there was a poor match between the MRI and final histopathological examination with an adjusted kappa of −0.08 [95% IC (−0.26; 0.09)].

For nodal invasion, there was a poor agreement between the MRI and final histopathological examination with an adjusted kappa of 0.16 [95% IC (−0.05; 0.38)]. On the 128 performed lymphadenectomies, we observed an under estimation of MRI on nodal metastases in 11.8% of cases (n = 15), an overestimation in 6.3% of cases (n = 8), and a concordance in 81.9% of cases (n = 105). There was no statistical difference between the frequency of cases with nodal metastases on MRI and final histological examination (*p* = 0.144) (Table 3).

Among women with concordance for nodal extension evaluation on MRI and histopathological examination 64% had a BMI higher than 35 kg/m^2^ for 57% for women with a discordance (*p* = 0.037).

There was no significant difference for the presence of other abnormalities on MRI on the uterus as fibroma, myoma, endometriosis, or internal material which could explain this mismatch on FIGO stage and myometrial invasion.

### 3.3. Concordance between Endometrial Biopsy and Final Histopathological Examination 

There was a moderate concordance on tumoral grade between the pretherapeutic biopsy and the final histopathological examination on excised tissue with an adjusted kappa of 0.52 [95% IC 0.42–0.62)].

Endometrial biopsy underestimated the tumor grade in 28.9% of cases (n = 50) (*p* < 0.001), overestimates the tumor grade in 6.9% of cases (n = 12), and we observed a concordance in 64.2% of cases (n = 111) (Table 4).

Moreover, diagnosis was performed by biopsy by Cornier pipelle in 81 cases (46.8%) and by hysteroscopy in 92 cases (53.2%). The biopsy performed by Cornier pipelle overestimated the tumor grade in 16.0% of cases (n = 13) and underestimated the tumor grade in 39.5% (n = 32) with a concordance in 58.0% (n = 47). Biopsies performed by hysteroscopy overestimated the tumor grade in 10.9% of cases (n = 10) and underestimated the tumor grade in 19.6% (n = 18) with a concordance in 69.6% of cases (n = 64). Biopsies by Cornier pipelle had more discordance than biopsy by hysteroscopy on grade evaluation (*p* = 0.005) (Table 5).

Furthermore, 52.6% of women with nodal metastases (n = 10) had a grade 3 tumor (*p* = 0.041), 42.1% (n = 8) had a grade 2 tumor and 5.3% (n = 1) had a grade 1 tumor. Twenty-five percent (n = 46) of women had LVSI on final histopathological examination and among these women 24% (n = 11) had nodal metastases and 60.1% (n = 28) had no nodal metastases (*p* = 0.004).

### 3.4. Indication of Lymphadenectomy

One hundred and twenty-eight lymphadenectomies were performed. Preoperatively, 66 women had an indication of pelvic and para-aortic lymphadenectomy. Among these 66 women, the indication of lymphadenectomy was maintained for 50 of them (43.1%) on the postoperative data. In total, of the 66 lymphadenectomies indicated on the preoperative data, 58 were performed (45.3%). Seventy were performed without any indication on the preoperative data (54.7%).

One hundred and twenty-eight lymphadenectomies were performed, 20 (11%) had both pelvic and para-aortic lymphadenectomies 108 women (59.7%) had a pelvic lymphadenectomy alone. Nineteen (10.4%) women had nodal metastases, 3 (1.6%) had both pelvic and para-aortic metastases, 15 (8.2%) had pelvic metastases alone, and 1 (0.5%) had para-aortic metastases alone.

For 6 women of these 19 women (31.6%) with nodal metastases, the lymphadenectomy was not indicated with the pretherapeutic data.

For 50 women (27.3%) lymphadenectomy was indicated on preoperative and postoperative data. For 16 women (8.7%) lymphadenectomy was indicated on preoperative data and not indicated on post-operative data. Among these women with an indication of lymphadenectomy, 58 (87.9%) had a lymphadenectomy and 13 (19.7%) had nodal metastases.

Lymphadenectomy was not indicated in both preoperative and postoperative data for 64 women (35.0%) and was not indicated on preoperative data but on postoperative data for 42 women (23.0%). Among these women with no indication of lymphadenectomy on preoperative data, 70 (66.1%) had a lymphadenectomy and 6 (8.6%) had nodal metastases (Table 6). All women with nodal metastases had an indication of lymphadenectomy on postoperative data.

Women with absence of para-aortic lymphadenectomy had a higher BMI than women with both pelvic and para-aortic lymphadenectomy (41.8 vs. 31.8, *p* = 0.012). We did not observe a significant difference in the ASA score (*p* = 0.333) or in the age (*p* = 0.85).

## 4. Discussion

Our study observed a significant under estimation on FIGO stage by MRI and endometrial biopsy.

Hysterectomy with bilateral salpingo-oophorectomy with or without pelvic and para-aortic lymphadenectomy is the primary treatment for endometrial cancer for stage I to III.

Several studies showed an absence of benefit in overall survival and recurrence-free survival for lymphadenectomy for stage I endometrial cancer compared to the risk of this surgical procedure [4] because of the absence of benefit of adjuvant radiotherapy on stage I endometrial cancer [18]. Other studies showed that systematic lymphadenectomy increased the overall survival especially with removal of more than ten lymph nodes but only for intermediate or high-risk endometrial cancer [19]. Nonetheless, knowledge of lymph node status can avoid adjuvant radiotherapy for women with no lymph node metastases [20]. This knowledge can also lead to adjuvant radiotherapy and chemotherapy for women with lymph node metastases which have a benefit on survival for women with lymph node metastases [9,10,21]. However, there is not a higher per-operative risk with pelvic and para-aortic lymphadenectomy, but there is a higher risk of surgery-related morbidity as lymphoedema and lymphocyst [22].

### 4.1. Concordance between MRI and Final Histopathological Examination

For myometrial infiltration, the sensitivity of MRI was 37% and specificity was 54% in our study. The under estimation by MRI on myometrial infiltration could be explained in literature by the presence of other uterine anomalies as myoma, this factor was not significantly in our study. Furthermore, the myometrial infiltration could be difficult to evaluate because of menopausal status. In post-menopause, we observed a thinning of myometrial tissue, which could complicate the estimation of myometrial infiltration [23].

Bi and al. showed, in a meta-analysis, higher sensitivity (79% vs. 37%) and higher specificity (81% vs. 54%) for evaluation of myometrial invasion by MRI [24]. These differences of MRI performance could be explained by multiple examinations performed in external centers by non-experts radiologists and by the age of our population. Indeed, they observed higher sensitivity (84% vs. 79%) and higher specificity (90% vs. 81%) for patients younger than 60 years old, or our population presented a median age of 66.1 years old (SD 10.1). Rei et al. also showed higher sensitivity (71% vs. 37%) and higher specificity (78% vs. 54%) for evaluation of myometrial invasion, but on a sample of 51 patients [25].

In fact, several MRI were not performed by expert center and every imaging center did not execute an axial section in T2, which evaluates the extension of the junctional area with the myometer and dynamic sequences in T1 with injection, which precise the depth of myometrial infiltration, especially for postmenopausal women [26]. The lack of some sequences in non-expert centers and the interpretation by non-expert radiologists highlighted the inter-observer variability.

Concerning our observation finding a higher BMI in women with a discordance between FIGO stage on MRI and on histopathological examination, we do not find to date any studies evaluating obesity as a factor influencing MRI images or finding this characteristic.

For lymph node evaluation, the sensitivity of MRI was 21% and the specificity was 93%. Lymph node metastases are determined by the augmentation of lymph node volume in MRI, but sensitivity was variable in literature from 17.0% to 70.6% [17]. Nonetheless, a lymph node was suspicious if it was higher than 10 mm in the majority of studies, but a lymph node could be metastatic with a size under 2 mm [27].

To confront this insufficient MRI performance, several studies evaluated the performance of PET-TDM in detection of nodal invasion. This examination showed a higher sensitivity (83.3% vs. 38.9%) and lower specificity (51.2% vs. 96.3%) than MRI in detection of pelvic metastases [28,29]. However, there was a false negative with PET-TDM for tumor size under 8 mm [30]. The use of machine learning also seems to be a tool that can improve the radiologist’s performance in evaluating myometrial infiltration [31].

In order to avoid surgical complication, several studies evaluated the performance of sentinel-node biopsy in endometrial cancer, especially in early stages. It showed a sensitivity from 84% to 97.2% and a negative predictive value from 97% to 100% with a lower morbidity [32,33,34,35]. This performance led to the recommendation of sentinel-node biopsy in stage I and II endometrial cancer in the United States of America in 2018 [36,37]. This method should be integrated in French recommendation.

### 4.2. Concordance between Endometrial Biopsy and Final Histopathological Examination

We found a moderate concordance on the grade between the pretherapeutic biopsy and the final histopathological examination on excised tissue with an adjusted kappa of 0.52 [95% IC 0.42–0.62)]. Several studies showed a wrong correlation on the grade between the endometrial biopsy and the final histopathological examination [38].

This discordance was mostly observed from grade 1 to grade 2. In fact, studies showed a better concordance between pre-operative samples and final histopathological examination for high grade in comparison to low and intermediate grades [39,40]. Although only grade 3 tumors have an indication of lymphadenectomy, this discordance between low and intermediate grades has no therapeutic consequence for non-conservative treatment.

The significant downgrading on endometrial biopsy could be explained by the different proportion of solid tumor and the heterogeneity of the tumor. Indeed, the proportion of nuclear atypia and differentiation is variable in the same tumor and depends on the sample. This bias could be reduced by multiple samples. Moreover, several samples were examined in external centers, which highlighted the inter-observer variability and poor reproducibility [41,42].

However, the pre-operative examination was more concordant with the final histopathological examination in case of sample realized by hysteroscopy [43]. There were some anesthetics and surgical adverse outcomes with hysteroscopy as hemorrhage (2.4%), uterine perforation (1.5%), and cervical laceration (1–11%) [44]. Some studies reported an increased risk of peritoneal dissemination after preoperative hysteroscopy, this risk was not increased and had no influence on the prognosis for stage I endometrial carcinoma [45].

### 4.3. Strength and Limitation

This was a monocentric and retrospective study. At least three generations of MRI machines were use in the reference center from 2009 to 2019, each generation leading to a better signal with less artefact. This could contribute to wrong estimation of local invasion on MRI. Several MRI and endometrial biopsies were examined in external centers which could be linked to a lack of experimentation in the interpretation.

The combined method by PET-MRI is on evaluation and primary observational studies showed at least an equivalence between PET-MRI and PET-TDM for the diagnosis of endometrial cancers and regional metastases [46,47]. Fernandez et al. showed a correct prediction of FIGO stage in 100% of the cases with three-dimensional transvaginal sonography but only on a sample of 20 women [48].

## 5. Conclusions

The pre-treatment assessment of endometrial cancer is not consistent with the results obtained on histopathological examination.

A systematic re-reading of the imaging must be carried out in an expert center by radiologists trained in the pathology, and this treatment must be integrated into a specific care management for endometrial cancer patients, in order to decide on the best therapeutic plan in multidisciplinary concertation meeting in expert centers.

The sentinel node technique, performed in groups considered low and intermediate risk, is developing and is necessary in view of the poor imaging performance.

## Figures and Tables

**Table 1 diagnostics-10-01045-t001:** Characteristics of population.

Age (years old) (Median, SD) (N = 183)	66.1 (10.1)
BMI (kg/m^2^) (Median, SD) (N = 183)	39 (14.5)
ASA score (n, %) (N = 170)	
1	29 (17.1)
2	76 (44.7)
3	61 (35.9)
4	4 (2.4)
Menopausal status (n, %) (N = 182)	
Premenopausal	6 (3.3)
Postmenopausal	176 (96.7)
Hormone replacement therapy (n, %) (N = 155)	28 (18.1)
Tamoxifen (n, %) (N = 182)	12 (6.6)
Parity (median, range) (N = 171)	2 (0–9)
Abnormal bleedings (n, %) (N = 183)	166 (90.7)
Diabetes (n, %) (N = 183)	40 (21.9)

BMI = body mass index, ASA = American Society of Anesthesiologists, n = number of women, N = number of women.

**Table 2 diagnostics-10-01045-t002:** Global concordance between MRI and final histopathological examination.

FIGO Stage		Final Histopathological Examination (n, %)				N (n, %)
		IA	IB	II	III–IV	
**MRI**	IA	36 (37.5)	36 (37.5)	11 (11.5)	13 (13.5)	96 (53.9)
	IB	21 (50.0)	11 (26.2)	5 (11.9)	5 (11.9)	42 (23.6)
	II	3 (17.6)	4 (23.5)	4 (23.5)	6 (35.3)	17 (9.6)
	III–IV	6 (26.1)	6 (26.1)	3 (13.0)	8 (34.8)	23 (12.9)
**N (n, %)**		66 (37.1)	57 (32.0)	23 (12.9)	32 (18.0)	178

FIGO = International Federation of Gynecology and Obstetrics, MRI = magnetic resonance imaging, n = number of women, N = total number of women. Adjusted Kappa of 0.12 [0.018; 0.24].

**Table 3 diagnostics-10-01045-t003:** Concordance between MRI and final histopathological examination on lymph node extension.

		Final Histopathological Examination (n, %)		
		N−	N+	
**MRI**	N−	101 (87.1)	15 (12.9)	116
	N+	8 (66.7)	4 (33.3)	12
		109	19	

MRI = magnetic resonance imaging, N− = no lymph node metastasis, N+ = lymph node metastasis, n = number of women. Adjusted kappa of 0.16 [−0.05; 0.38]. Sensitivity: 4/19 = 21%, specificity: 101/109 = 92.7%.

**Table 4 diagnostics-10-01045-t004:** Concordance between endometrial biopsy and final histopathological examination on tumor grade.

Tumor Grade		Final Histopathological Examination (n, %)			N (n, %)
		1	2	3	
**Endometrial biopsy**	1	37 (49.3)	35 (46.7)	3 (4.0)	75 (43.4)
	2	5 (7.6)	49 (74.2)	12 (18.2)	66 (38.2)
	3	1 (3.1)	6 (18.8)	25 (78.1)	32 (18.5)
**N (n, %)**		43 (24.9)	90 (52.0)	40 (23.1)	173

n = number of women, N = number of women. Adjusted kappa of 0.52 [0.42; 0.62].

**Table 5 diagnostics-10-01045-t005:** Concordance on tumor grade MRI and final histopathological examination on endometrial biopsy by Cornier pipelle and hysteroscopy.

Tumor Grade		Final Histopathological Examination (n, %)			N (n, %)
		1	2	3	
**Cornier pipelle (n, %)**	1	13 (34.2)	22 (57.9)	3 (7.9)	38
	2	2 (6.5)	22 (71.0)	7 (22.6)	31
	3	0	0	12 (100)	12
**Hysteroscopy (n, %)**	1	24 (64.9)	13 (35.1)	0	37
	2	3 (8.6)	27 (77.1)	5 (14.3)	35
	3	1 (5.0)	6 (30.0)	13 (65.0)	20
**N (n, %)**		43 (24.9)	90 (52.0)	40 (23.1)	173

n = number of women, N = number of women.

**Table 6 diagnostics-10-01045-t006:** Indication of lymphadenectomy and nodal metastases.

	Indication of Lymphadenectomy on Postoperative Data (n/N)				
**Indication of lymphadenectomy on preoperative data (n/N)**		**Indicated**	**Not Indicated**	**Performed**	**Nodal Metastasis**
	Indicated	50/66	16/66	58/66	13/58
	Not indicated	42/106	64/106	70/106	6/70

N, n = number of women.

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
