# Peer review of "Evaluation of Pre-Therapeutic Assessment in Endometrial Cancer Staging"

_diagnostics, 2020, doi:10.3390/diagnostics10121045_

Round 1
Reviewer 1 Report
In this manuscript, Bouche et al. report a retrospective study evaluating the concordance between pre-operative MRI and biopsy and final surgical histopathology. Whilst the study is of interest, the authors have not discussed related studies. The manuscript will benefit from editing by a native English speaker.
“Sensibility” is used throughout the manuscript, do the authors mean “sensitivity”?
Abstract:
Conclusion states “in order to consider an endometrium course”, could the authors please explain what is meant by this statement and re-word accordingly.
Introduction:
The Introduction refers several times to lymph node or other “injury” – are the authors referring to infiltration?
Materials and Methods:
Does “hormone status” refer to menopausal status?
Please note that “histolopathological” is not a word, this should be “histopathological”.
Need to specify whether histopathology and MRI images were reviewed by >1 person and were they blinded to outcomes?
Need to insert details of how MRI images were assessed.
Results:
Table 1: What do the numbers in brackets for age and BMI refer to? Define ASA. Also, the total number of patients appears to be consistent for all characteristics assessed, e.g. the number of patients with ASA score is 170, yet the number of patients with hormonal/menopausal status is 182. Why are these numbers inconsistent and how many patients were included in the other groups?
Table 2: Data would be clearer and easier to interpret if the percentages in brackets represented the proportion of cases by stage. For example, there are 96 cases that are stage IA as determined by MRI. Of these 96 cases, 36 were also stage IA (37.5%) on final histopathology, 36 were stage IB (37.5%), 11 were stage II (11.5%) and 13 were stage III-IV (13.5%).
Does concordance between MRI and final histopathology vary by stage? The authors mention grade, but Table 2 refers to stage.
Table 3: Why is data only presented for 127 patients? Again, data would be clearer and easier to interpret if the percentages in brackets represented the proportion of cases by nodal status, i.e. 115 cases were N- on MRI. Of these 115 cases, 100 (87%) were also N- on final histopathology, whilst 15 (13%) were N+.
Table 4: Data would be clearer and easier to interpret if the percentages in brackets represented the proportion of cases by grade. See comments for Tables 2 and 3.
Data on concordance between diagnostic method (pipelle or hysteroscopy) and final histopathology would be much clearer if presented in a Table.
The number of patients who underwent lymphadenectomy is unclear (page 10, first and second paragraphs) – please clarify numbers.
Table 5: Please clearly explain what the numbers in this table mean and insert percentages as in other tables to make interpretation clearer and easier.
Page 10, last paragraph: “abstention of para-aortic lymphadenectomy”, do the authors mean “absence of para-aortic lymphadenectomy”?
Discussion:
This study observed a significant underestimation of stage by MRI. Other studies have shown higher concordance between clinical and histopathological staging, especially in stage IA cases (e.g. 2017 Korean Gynecologic Oncology Group study by Lee et al.) – the authors need to reference related studies and comment on why their results are so different.
Inter-observer variability between radiologists is discussed, but this has not been mentioned elsewhere in the manuscript. Details on how images were acquired and interpreted need to be added to the Methods section and relevant information added to the Results section.
Need to insert conclusions on MRI and nodal status.
Discordance between grades 1 and 2 has therapeutic consequences if patients are being considered for conservative management (e.g. progesterone therapy for fertility-sparing treatment).
Inter-observer variability between pathologists is also discussed but not mentioned elsewhere in the manuscript – please expand.
Need to comment on MRI and BMI as this was mentioned in the Results section.
References:
References need to be updated with more recent studies cited.
Author Response
In this manuscript, Bouche et al. report a retrospective study evaluating the concordance between pre-operative MRI and biopsy and final surgical histopathology. Whilst the study is of interest, the authors have not discussed related studies. The manuscript will benefit from editing by a native English speaker.
Thank you for your attentive review and your many comments to improve the quality of our work.
Concerning our English mistakes: unfortunately, the time limit imposed for the corrections did not allow us to make an official correction. We have corrected all the errors you pointed out to us and those we noticed, a correction can be made in a second step if necessary.
Abstract:
Conclusion states “in order to consider an endometrium course”, could the authors please explain what is meant by this statement and re-word accordingly.
Thank you for your comment. We have developed and rephrased what we meant by endometrium pathway.
Introduction:
The Introduction refers several times to lymph node or other “injury” – are the authors referring to infiltration?
Thank you for your comment. We have corrected “injury” by “infiltration”.
Materials and Methods:
Does “hormone status” refer to menopausal status?
Thank you for your comment. We have corrected “hormonal status” by “menopausal status”.
Please note that “histolopathological” is not a word, this should be “histopathological”.
Need to specify whether histopathology and MRI images were reviewed by >1 person and were they blinded to outcomes?
Need to insert details of how MRI images were assessed.
Thank you for your comment. We have specified in the materials and methods section the MRI protocol and the absence of proofreading. Indeed, the MRIs were mostly performed in external centers and some in expert centers and interpreted by a single radiologist.
Results:
Table 1: What do the numbers in brackets for age and BMI refer to? Define ASA. Also, the total number of patients appears to be consistent for all characteristics assessed, e.g. the number of patients with ASA score is 170, yet the number of patients with hormonal/menopausal status is 182. Why are these numbers inconsistent and how many patients were included in the other groups?
Thank you for your comment. The numbers in brackets refer to the standard deviation, we have added the information in the table. The ASA score has also been defined in the material and method section. The numbers vary because the information was collected whenever it was available in the chart, which unfortunately was not the case for all of our patients.
Table 2: Data would be clearer and easier to interpret if the percentages in brackets represented the proportion of cases by stage. For example, there are 96 cases that are stage IA as determined by MRI. Of these 96 cases, 36 were also stage IA (37.5%) on final histopathology, 36 were stage IB (37.5%), 11 were stage II (11.5%) and 13 were stage III-IV (13.5%).
Does concordance between MRI and final histopathology vary by stage? The authors mention grade, but Table 2 refers to stage.
Thank you for your comment. We have presented the figures following your advice. The sentence mentioning the variation by grade was confusing, so we have chosen to delete it.
Table 3: Why is data only presented for 127 patients? Again, data would be clearer and easier to interpret if the percentages in brackets represented the proportion of cases by nodal status, i.e. 115 cases were N- on MRI. Of these 115 cases, 100 (87%) were also N- on final histopathology, whilst 15 (13%) were N+.
Thank you for your comment. The results are presented for 128 patients because of the 183 patients operated on, only 128 had a lymphadenectomy. We also followed your advice for the presentation of the figures.
Table 4: Data would be clearer and easier to interpret if the percentages in brackets represented the proportion of cases by grade. See comments for Tables 2 and 3.
Thank you for your comment. We also followed your recommendations for the presentation of the figures.
Data on concordance between diagnostic method (pipelle or hysteroscopy) and final histopathology would be much clearer if presented in a Table.
Thank you for your comment. We have followed your recommendation and created a new table (Table 5) representing the concordance on tumor grade between diagnostic method (pipelle or hysteroscopy) and final histopathology.
The number of patients who underwent lymphadenectomy is unclear (page 10, first and second paragraphs) – please clarify numbers.
Table 5: Please clearly explain what the numbers in this table mean and insert percentages as in other tables to make interpretation clearer and easier.
Thank you for your comment. We have specified the number of patients who have had a lymphadenectomy at the beginning of the "indication for lymphadenectomy" section and tried to make the section clearer in order to better understand Table 6 (formerly Table 5).
We had initially chosen the presentation in percentage form as you suggested, however this form seemed less clear to us than the current form chosen. After discussion with the authors, we did not find a clearer form.
Page 10, last paragraph: “abstention of para-aortic lymphadenectomy”, do the authors mean “absence of para-aortic lymphadenectomy”?
Thank you for your comment. We have corrected “abstention” by “absence”.
Discussion:
This study observed a significant underestimation of stage by MRI. Other studies have shown higher concordance between clinical and histopathological staging, especially in stage IA cases (e.g. 2017 Korean Gynecologic Oncology Group study by Lee et al.) – the authors need to reference related studies and comment on why their results are so different.
Thank you for your comment. We have further compared our results with those of other studies. However, we looked for the reference you suggested (2017 Korean Gynecologic Oncology Group study by Lee et al.) and did not find any comparison between the estimation of FIGO stage by imaging and histology.
Inter-observer variability between radiologists is discussed, but this has not been mentioned elsewhere in the manuscript. Details on how images were acquired and interpreted need to be added to the Methods section and relevant information added to the Results section.
Inter-observer variability between pathologists is also discussed but not mentioned elsewhere in the manuscript – please expand.
Concerning the inter-observer variability between radiologists and pathologists, thank you for your comment. We have specified that MRI and biopsies were performed in different centers in the materials and methods section. However, we have not collected the number of examinations performed in external centers and those performed in expert centers. Our team of expert radiologists is currently doing this work on our population.
Need to insert conclusions on MRI and nodal status.
Thank you for your comment. We have added a conclusion on MRI and lymph node status.
Discordance between grades 1 and 2 has therapeutic consequences if patients are being considered for conservative management (e.g. progesterone therapy for fertility-sparing treatment).
Thank you for your comment. We have specified that the discordance between grades 1 and 2 has no consequence in the case of non-conservative treatment.
Need to comment on MRI and BMI as this was mentioned in the Results section.
Thank you for your comment. Despite an influence of BMI in the concordance of FIGO stage observed in our study, we did not find this notion in the literature. This information has been added to the discussion section.
References:
References need to be updated with more recent studies cited.
Thank you for your comment. As you suggested, we have updated our bibliography.
Reviewer 2 Report
As is stated in the Introduction, “Preoperative assessment of the risk of lymph node metastasis is critical in the management of Type I endometrial cancer.” A retrospective analysis, comparing MRI prediction with actual pathologic analysis (and FIGO staging), is then certainly meaningful.
It is well-understood that tumor grade correlates with the extent of myometrial invasion, which in turn correlates with lymph nodal spread of disease. A pre-operative assessment of myometrial and lymph node invasion can assist surgeons in deciding on whether to perform pelvic and/or para-aortic lymphadenectomy. It is fair to make this same comparison with the reliance on intra-operative Frozen Section or Sentinel Lymph Node mapping.
The statistical analyses appear to be appropriate, as were the conclusions reached, and publication of these findings is certainly warranted. My only comments relative to this investigation are that this same sort of comparison should be undertaken with other imaging tools, even if such other tools are not currently used in practice, considering its conclusion, that “the pre-treatment assessment of endometrial cancer [with MRI} is not consistent with the results obtained on histopathological examination.” Another pre-operative imaging tool to consider for this sort of comparison is Three-Dimensional Transvaginal Sonography. However, while 3DTVS can identify the depth of invasion, it is not useful for identifying lymph node spread. An example of such a comparison is listed below, revealing my reviewer bias.
Fernandez CM, Levine EM, Dini M, Bannon K, Butler S, Locher S: Predictive value of three-dimensional transvaginal sonography for staging of endometrial carcinoma. J Diagn Med Sonogr 2018;34(5):496-500.
Author Response
My only comments relative to this investigation are that this same sort of comparison should be undertaken with other imaging tools, even if such other tools are not currently used in practice, considering its conclusion, that “the pre-treatment assessment of endometrial cancer [with MRI} is not consistent with the results obtained on histopathological examination.” Another pre-operative imaging tool to consider for this sort of comparison is Three-Dimensional Transvaginal Sonography. However, while 3DTVS can identify the depth of invasion, it is not useful for identifying lymph node spread. An example of such a comparison is listed below, revealing my reviewer bias.
Fernandez CM, Levine EM, Dini M, Bannon K, Butler S, Locher S: Predictive value of three-dimensional transvaginal sonography for staging of endometrial carcinoma. J Diagn Med Sonogr 2018;34(5):496-500.
Thank you for your attentive review and your comments to improve the quality of our work.
Following your comment, we have added in the discussion section, the notion of evaluation by 3D ultrasound by quoting the reference you suggested.
Round 2
Reviewer 1 Report
The authors have addressed most comments; however, a few things still need clarifying.
This manuscript requires English correction before it can be published. Some examples of statements requiring clarification or correction:
Line 61: “determines the indication” – indication of what? Treatment or something else?
Line 157: “comparatively” should be “compared to”.
Line 193: delete “realised”.
Lines 194 and 196: “realised” should be “performed”.
Line 207: “metastase” should be “metastases”.
Line 232: “on” should be “in” in both cases.
Line 242: “specially” should be “especially”.
Lines 252 and 274: should be “sensitivity of MRI was X% and specificity was X%”.
Line 268: “which precise the myometrial infiltration” – this sentence is unclear, please clarify.
Line 316: “exanimated” should be “examined”.
Abstract:
The sentence “in order to consider an endometrium pathway” still does not make any sense – what message are the authors trying to convey?
Results:
Data in Table 1 should be for the same number of patients for each characteristic (n=183). For transparency, the number of patients for whom data was not available for each characteristic should clearly be stated in the table. For example, ASA status was not available for 13 patients and this should be clearly stated within the table.
Table 6 indicates 66 lymphadenectomies were performed without indication on pre-operative data, however line 213 states 70 were performed. The table also indicates a total of 124 lymphadenectomies were performed, but the text states 128. Could the authors please clarify these inconsistencies?
Discussion:
The authors refer to “downstaging” by MRI multiple times in the discussion. Downstaging implies that stage was overestimated, however, results here indicate that stage was underestimated by MRI. Please clarify or correct.
Author Response
This manuscript requires English correction before it can be published
Thank you for your comments. An official translation is currently in progress.
Abstract:
The sentence “in order to consider an endometrium pathway” still does not make any sense – what message are the authors trying to convey?
Thank you for your comment. We reformulated by "specific care management" and developed our idea.
Results:
Data in Table 1 should be for the same number of patients for each characteristic (n=183). For transparency, the number of patients for whom data was not available for each characteristic should clearly be stated in the table. For example, ASA status was not available for 13 patients.
Thank you for your comment. We have specified the number of patients for whom data were available. s and this should be clearly stated within the table.
Table 6 indicates 66 lymphadenectomies were performed without indication on pre-operative data, however line 213 states 70 were performed. The table also indicates a total of 124 lymphadenectomies were performed, but the text states 128. Could the authors please clarify these inconsistencies?
Thank you for your comment. We have corrected these inconsistencies.
Discussion:
The authors refer to “downstaging” by MRI multiple times in the discussion. Downstaging implies that stage was overestimated, however, results here indicate that stage was underestimated by MRI. Please clarify or correct.
Thank you for your comment. We have corrected this vocabulary error.